# Microwave-Induced Behavior and Digestive Properties of the Lotus Seed Starch—Chlorogenic Acid Complex

**DOI:** 10.3390/foods12132506

**Published:** 2023-06-28

**Authors:** Xiangfu Jiang, Jianyi Wang, Lanxin Li, Baodong Zheng, Shuyi Zheng, Xu Lu

**Affiliations:** 1College of Food Science, Fujian Agriculture and Forestry University, Fuzhou 350002, China; jxf020@126.com (X.J.); joneeg@163.com (J.W.); lanxin1559@163.com (L.L.); zbdfst@163.com (B.Z.); zshuyi2000@163.com (S.Z.); 2Fujian Provincial Key Laboratory of Quality Science and Processing Technology in Special Starch, Fujian Agriculture and Forestry University, Fuzhou 350002, China; 3China-Ireland International Cooperation Centre for Food Material Science and Structure Design, Fujian Agriculture and Forestry University, Fuzhou 350002, China

**Keywords:** microwave, dielectric properties, starch gelatinization, starch-polyphenol complex

## Abstract

The effect of chlorogenic acid (CA) on the dielectric response of lotus seed starch (LS) after microwave treatment, the behavior and digestive characteristics of the resulting starch/chlorogenic acid complex (LS-CA) at different degrees of gelatinization and the inhibition of α-amylase by chlorogenic acid were investigated. The variation in dielectric loss factor, ε″, and dielectric loss tangent, tanδε, of the microwave thermal conversion indicated that LS-CA had a more efficient microwave-energy-to-thermal-energy conversion efficiency than LS. This gelatinized LS-CA to a greater extent at any given temperature between 65 and 85 °C than LS, and it accelerated the degradation of the starch crystalline structure. The greater disruption of the crystal structure decreased the bound water content and increased the thermal stability of LS-CA compared to LS. The simulated in vitro digestion found that the presence of the LS-CA complex improved the slow-digestion property of lotus seed starch by increasing its content of resistant and slowly digested starch. In addition, the release of chlorogenic acid during α-amylase hydrolysis further slowed starch digestion by inhibiting α-amylase activity. These findings provide a foundation for understanding the correlation between the complex behavior and digestive properties of naturally polyphenol-rich, starch-based foods, such as LS, under microwave treatment, which will facilitate the development of starch-based foods with tailored digestion rates, lower final degrees of hydrolysis and glycemic indices.

## 1. Introduction

Microwaves are electromagnetic waves with a frequency range of 300 MHz–300 GHz and a wavelength range of 1 mm–1 m. Microwave treatment has become an efficient and convenient method of thermomechanical food processing and is widely used in both home cooking and the food industry [1]. Unlike conventional heating methods, microwave heating is a multi-physical field phenomenon based on the dielectric effect (non-ionizing radiation generated by a high frequency alternating magnetic field causes oscillation and friction of polar molecules within the material, thus resulting in energy transformation), which has the advantages of short heating times, high penetration, and high energy efficiency [2]. This makes microwaves an efficient processing method, which is widely used in the processing and production of starch-based foods [3]. Under normal atmospheric pressure conditions, the 10–20% of water contained in starch absorbs most of the microwave energy (water molecules are more polar than starch molecules), which is transferred to the starch as heat [4]. The mechanism of action of microwave heating of starch is as follows: (1) the dielectric relaxation of water molecules results in heating of the water molecules in the starch; (2) the evaporation/boiling of water leads to a rapid increase in temperature and pressure inside the starch granules; (3) the high pressure forces the granules to swell, starting from the highest temperature point and spreading throughout the granule; (4) swelling; this results in the disruption and gelatinization of the granules. Water molecules strongly influence the dielectric properties of starch, and these properties are the key to the efficient conversion of microwave energy to heat.

The microwave cooking of starch-based foods is mostly accompanied by starch pasting/gelatinization. Water molecules are the main mediators of starch gelatinization, which enables starch granules to absorb water, swell and dissolve at high temperatures, forming a paste/gel [5,6]. However, the microwave gelatinization of starch is influenced by other factors, especially the presence of phenolic compounds. During processing, such small molecules can interact with starch to form complexes that improve its gelatinization properties and enhance the quality of starch-based products. Phenolic compounds can bind to other polar molecules, such as water molecules, and trigger interfacial polarization, as well as local relaxation, thereby affecting the dielectric response of target substrates, enhancing its more efficient use of microwave energy [7,8]. Past studies on the interactions between starch and polyphenolic compounds during microwave treatment mostly focused on starch which had been gelatinized, or the subsequent starch aging/retrogradation, whereas there are relatively few studies on the influence of polyphenolic compounds on starch pasting during microwave treatment [9,10,11].

Lotus seed is the mature seed of *Nelumbo nubifera* Gaertn, which has a long historical tradition of consumption and medicinal use in Asia (China, Japan, Vietnam and other Southeast Asian countries), because of its effects of lowering blood pressure, regulating heart rate, relieving anxiety and improving sleep [12]. As a typical highly starchy food, the starch content of lotus seeds exceeds 60% of their total dry basis weight. Therefore, the property changes in lotus seed starch during processing play an important role in influencing the processing characteristics of lotus seeds. Pasteurization is the key factor affecting the starch properties, especially the combination of endogenous compounds (e.g., polyphenols) released from lotus seed cells during processing, whereby pasteurization will significantly change the pasteurization properties of starch (increased crystal disintegration, advancement of pasteurization point, etc.). Chlorogenic acid, the most abundant endogenous phenolic substance in lotus seeds (66%), has a variety of bioactive functions such as antibacterial, antiviral and mutation inhibition [13,14]. In lotus seeds, it not only directly enters the straight-chain starch as a small molecule and reacts with the starch hydroxyl group through the highly active hydroxyl group on the surface, but it also modifies the starch by combining with the branched chain starch. The interaction between chlorogenic acid and starch can be regulated to a certain extent by processing methods, compared to other processing methods (hydrothermal method, high-pressure homogenization, etc.). Microwave is an efficient and convenient processing method, wherein the thermal effect of microwave can quickly heat up the starch molecules to achieve the paste effect and promote the starch–polyphenol complex, thus leading to the formation of resistant starch for the prevention and control of type II diabetes, colon (rectal) cancer and many chronic diseases and cardiovascular and cerebrovascular diseases [15,16,17].

In this study, lotus seed starch–chlorogenic acid complexes were prepared by microwave treatment, and their physicochemical and digestive properties were determined, in different gelatinization states, to clarify the relationship between them. The findings will help to establish a theoretical basis for the application of microwave processing to starch-based foods.

## 2. Materials and Methods

### 2.1. Materials

Fresh lotus seeds (Green Field Food Co. Ltd., Fujian, China) were mixed with distilled water at a 1:3 ratio, disrupted in a tissue masher for 1 min (DS-200, Changzhou Xiangtian Experimental Instrument Factory, Changzhou, China), passed through a 120-mesh filter and then allowed to settle for 24 h. The supernatant was discarded, and the sediment was washed with deionized water until there were no visible impurities on the surface. Finally, the sediment was stirred for 10 h in 95% *v/v* aqueous ethanol to remove polyphenols and lipids. The resulting lotus seed starch (LS) contained 9.55% moisture, 0.30% ash, 0.33% protein and 0.27% lipids, as determined by Chinese national standard methods. Chlorogenic acid (CA) was from Shanghai Yuanye Bio-technology Co., Ltd., Shanghai, China.

### 2.2. Dielectric Measurements

A dielectric data acquisition system (E5071C, Agilent Corporation, Santa Clara, CA, USA) consisting of a vector network analyzer, a dielectric probe and a multiplexed data logger was constructed as described previously [18]. A three-point calibration of “air, water, and metal short” was performed prior to measurement, and the water was adjusted to the appropriate temperature for each calibration to ensure the accuracy of the measurement. The samples were placed in glass beakers and wrapped in polystyrene during the test to reduce heat loss. Each measurement was repeated three times in parallel, averaged, and the dielectric constant, ε′, dielectric loss factor, ε″, and dielectric loss tangent, tanδε, were calculated directly using the instrument software (Version: 18.0.19012401).

### 2.3. Microwave Treatment of Lotus Seed Starch

An 8% *w/v* starch suspension in distilled water, with 0 (control), 1, 3, 5, 7, or 9% *w*/*w* chlorogenic acid (based on the dry weight of starch) was treated in a microwave extractor (XH300B, Beijing Xianghu Science and Technology Development Co. Ltd., Beijing, China), equipped with a water circulation and temperature monitoring system and heated (210 W) to 15 different final temperatures in the range 30−100 °C, with an interval of 5 °C between each temperature, to simulate the starch pasting process. At each set temperature, samples were removed and washed (×3) with 50% *v/v* aqueous ethanol to remove unbound CA from the LS, freeze-dried, crushed using a pulverizer and filtered through a 100 mesh screen to obtain powdered starch test samples.

### 2.4. Determination of Relative Degree of Complexation via the Quantification of Chlorogenic Acid with the Total Phenolic Content assay

The degree of complexation was determined as described previously [19], with minor modifications. Starch sample (10 mg) was accurately weighed and dissolved in dimethyl sulfoxide (10 mL), magnetically stirred for 30 min and then centrifuged (3000× *g* for 10 min). Supernatant (0.5 mL) was mixed with Folin-Ciocalteu reagent (1 mL), deionized water was added to a total volume of 4 mL, and then Na_2_CO_3_ solution (1 mL, 8%, *w*/*v*) was added, and the mixture was left to stand in the dark for 20 min. The absorbance was measured at 760 nm (UV-VIS Spectrophotometer, Shimadzu Corp., Kyoto, Japan) and the chlorogenic acid content calculated from a standard curve of gallic acid (concentration range 0–200 μg/mL, R^2^ = 0.9994, *n* = 6).

### 2.5. X-ray Diffraction (XRD) Analysis

XRD was performed as described previously [20], with minor modifications. A monochromatic copper target (Cu-Kα) ray (λ = 0.1789 nm) was used with a tube voltage of 40 kV, a tube current of 35 mA and a scan angle (2θ) range of 5° to 40° in steps of 0.05° (Bruker AXS, Bruker Corporation, Karlsruhe, Germany). The relative crystallinity was calculated as follows:$$RC\;(\mathrm{relative}\;\mathrm{crystallinity})=\frac{A_{c}}{A_{c}+A_{a}^{}}\times 100\%$$
where A_c_ refers to the area of the crystalline region, which is the area of the peaks in the diffraction pattern, and A_a_ refers to the area of the amorphous “hump”, which is the area of the region enclosed by the diffusion curve at the bottom of the diffraction pattern.

### 2.6. Thermogravimetric Analysis

The thermal properties of the starch samples (i.e., the temperatures of the different stages of thermal decomposition) were analyzed via a thermogravimetric analyzer (TGA8000, PerkinElmer Corporation, Waltham, MA, USA). The samples (5 mg) were heated in the temperature range 30−800 °C under a nitrogen atmosphere (50 mL/min). The heating rate was 10 °C/min.

### 2.7. Determination of Starch Swelling Power and Solubility

The starch sample was mixed with 20 volumes of deionized water and shaken in a reciprocating shaker (160 rpm) at room temperature (25 °C) for 1 h, and then, the mixture was centrifuged at 5000× *g* for 10 min. The supernatant was dried to constant weight and the weights of the dried residue and the centrifugation pellet recorded; the swelling power and solubility were calculated as follows:$$\;\mathrm{Swelling}\;\mathrm{force}\;(\mathrm{SP})\;=\frac{\;(\mathrm{Pellet}\;\mathrm{weight})}{\left(\mathrm{Sample}\;\mathrm{weight})-(\mathrm{Dried}\;\mathrm{residue}\;\mathrm{weight}\right)}$$
$$\;\mathrm{Solubility}\;(\%S)=\frac{\left(\mathrm{Dried}\;\mathrm{residue}\;\mathrm{weight}\right)}{\left(\mathrm{Sample}\;\mathrm{weight}\right)}\times 100\%$$

### 2.8. In Vitro Digestion

#### 2.8.1. Digestibility Determination

Digestibility was determined as described previously [21], with minor modifications. Starch sample (200 mg) was dispersed in sodium acetate buffer (15 mL, 0.1 M, pH 5.2). A freshly prepared enzyme solution (10 mL, 290 unit/mL α-amylase and 15 unit/mL amyloglucosidase) was added, and the mixture was shaken (SHZ-82 orbital shaker, Changzhou Ronghua Instrument Manufacturing Co., Ltd., Changzhou, China) to simulate an in vitro digestion environment (37 °C, 170 rpm). A sample (0.5 mL) of reaction mixture was removed at 0, 5, 10, 15, 20, 30, 40, 50, 60, 80, 100, 120, 150 and 180 min, and ethanol (2 mL) was added to inactivate the enzymes. The glucose content of the system was determined using a Megazyme Glucose Assay Kit (Megazyme Corporation, Wicklow, Ireland). The starch hydrolysis rate was determined from the production of glucose and the contents of rapidly digested starch (RDS, starch digested within 20 min), slowly digested starch (SDS, starch digested in the period of 20–120 min) and resistant starch (RS, starch undigested after 120 min). The corresponding equations were as follows:$$RDS\;(\%)=\frac{\left(G_{20}-G_{0}\right)\times 0.9}{TS}\times 100\%$$
$$SDS\;(\%)=\frac{\left(G_{120}-G_{20}\right)\times 0.9}{TS}\times 100\%$$
$$RS\;(\%)=\frac{\left(TS-G_{120}\right)\times 0.9}{TS}\times 100\%$$
where G(t) refers to the amount of glucose produced at digestion time t. TS is the initial amount of starch.

#### 2.8.2. Determination of Chlorogenic Acid (CA) Released during Digestion

The amount of chlorogenic acid released during digestion was determined by the method of Section 2.4.

### 2.9. Determination of α-Amylase Inhibition and the Type of Inhibition by CA

Amylase inhibition was determined as described previously [22,23] with minor modifications. Chlorogenic acid solution (1 mL; 0, 0.08, 0.17, 0.5, 1, 1.5 or 2 mg/mL) and α-amylase were combined and mixed in a water bath (HH-4, Guohua Electric Co., Ltd., China) at 37 °C for 20 min, to a final enzyme activity of 0.35 U/mL). Starch solution (1 mL, 1% *w*/*v*) was added and incubated for 10 min, and then dinitrosalicylic acid reagent (1 mL) was added and mixed, heated in a boiling water bath for 5 min and then cooled to room temperature. The volume was adjusted to 15 mL and the absorbance measured at 540 nm. The percentage inhibition was calculated as follows:$$\;\mathrm{Inhibition}\;(\%)=\left(A_{b}-A_{s}\right)\times 100/A_{b}$$
where: A_b_ = A_control_ − A_control blank_; and A_s_ = A_sample_ − A_sample blank_.

The Michaelis–Menten kinetics and the type of inhibition were determined from Lineweaver–Burk plots [24,25]. The concentrations of chlorogenic acid (0, 1, 1.5, 2 mg/mL) and starch solution (1, 1.25, 2.50, 5.00, 10.00 mg/mL) were varied, and 1/v was plotted against 1/[S]. The K_m_ and V_max_ were calculated as follows:$$\frac{1}{V}=\frac{K_{m}+[S]}{V_{max}[S]}=\frac{K_{m}}{V_{max}}\frac{1}{\left[S\right]}+\frac{1}{V_{max}}$$
where 1/V_max_ corresponds to the intercept on the *Y*-axis, and −1/K_m_ corresponds to the intercept on the *X*-axis.

### 2.10. Data Processing and Analysis

SPSS 17.0 and Origin Pro 7.5 software were used for data processing and correlation analysis. Each experiment was performed three times in parallel, and the results are expressed as the mean ± standard deviation (SD), with a significance level of *p* < 0.05.

## 3. Results and Discussion

### 3.1. Effect of Chlorogenic Acid on the Dielectric Properties of Lotus Seed Starch

The dielectric response, a key determinant of the propagation of microwave energy in materials, is characterized by the dielectric constant (ε′, the ability of a substance to absorb and store electrical energy), the dielectric loss factor (ε″, the ability of a substance to convert electrical energy into thermal energy) and the dielectric loss tangent (tanδε = ε″/ε′, the degree to which a substance converts absorbed electromagnetic energy into thermal energy) [26]. The effects of different chlorogenic acid (CA) concentrations on the dielectric parameters of lotus seed starch (LS) at 2.45 GHz (household microwave frequency), at room temperature, were determined (Table 1). The ε′ of LS-CA gradually decreased, and ε″ and tanδε gradually increased with an increasing CA concentration. LS-CA appeared to be an inhomogeneous medium in the microwave field, and differences in polarization of its different dispersed phases result in a potential difference at the phase interface. This induced charge separation to form “micro-capacitances”, resulting in a Maxwell–Wagner–Sillars polarization, over distances many times the molecular size; the polarized molecules at the phase junctions undergo a corresponding dielectric relaxation phenomenon [27]. Therefore, interactions with the polar CA molecules affect the dielectric response of the starch molecules during microwave treatment, compared with that of LS. However, as indicated by the negligible changes in dielectric parameters above 5% CA (Table 1), the polarizing effect essentially plateaus above 5% CA, so this concentration was used for all subsequent experiments.

The microwave-assisted gelatinization of starch involved molecules within the starch/water system absorbing microwave energy, which was converted into thermal energy by the polarization/relaxation process described above; the starch granules swelled and ruptured, releasing starch molecules into the water phase to form a gel/paste. The ε′ of LS decreased with a temperature increasing from 30 to 100 °C, and that of LS-CA decreased similarly but was lower at all temperatures (Figure 1A).

The ability of the starch system to absorb microwave energy depends mainly on the number of polarizable dipoles (free water molecules) per unit volume of the starch suspension [6]. In the solid state of LS in the granules, the water molecules are mostly immobilized and only weakly polarizable, so they do not absorb microwave energy efficiently. As the temperature increases, the starch granules undergo a phase change (swelling and rupture) and release starch and molecules, which enables the water molecules, previously immobilized inside the granules, to become more mobile, resulting in a gradual decrease in the overall dielectric constant of the starch. In the LS-CA system, CA, as a caffeoyl quinic acid derivative, can interact extensively with water molecules via hydrogen bonding. These additional hydrogen bonds affect the polarizability of water molecules, making it more difficult for them to absorb microwave energy and enhancing the constraining effect of starch granules and free starch molecules.

The ε″ and tanδε decreased from 30 to 75 °C and then increased from 75 to 100 °C (Figure 1B,C). There was a close relationship between the energy conversion pathway of molecules during microwave treatment and the relaxation of polar molecules in the system [28]; usually, the relaxation time of polar molecules decreased with an increasing temperature. However, in the starch suspension system, at the gelatinization temperature (75 °C), the starch granules had already swollen and ruptured, so the starch system was in a liquid crystal-like, dispersed state [29], in which polar molecules were unable to respond instantly to the change in electromagnetic field direction, resulting in hysteresis. This would promote more microwave energy absorption during the polarization phase to be dissipated in the form of thermal energy, manifesting as the increases in ε″ and tanδε above the pasting temperature. In contrast, water molecules in LS-CA were less polarizable and absorbed microwave energy more efficiently, reducing the transition temperature to about 70 °C, suggesting that the constraining effect of CA could also enhance the polarization hysteresis of water molecules. Therefore, the LS-CA hybrid system had higher microwave energy—thermal energy conversion efficiency and greater starch gelatinization at any given temperature.

### 3.2. Extent of Complexation of Chlorogenic Acid in LS-CA with Temperature

The complexation of small molecules with starch is influenced by various factors, such as the degree of starch gelatinization, the temperature and competitive binding between molecules [15]. After the LS-CA system was washed with ethanol, the free CA was removed, and only the CA bound to LS was retained. As the temperature increased during microwave treatment, the amount of bound/complexed CA initially increased gradually, then it increased rapidly from the gelatinization temperature (~70 °C) to a maximum at 85 °C (28.57 ± 0.52 mg/g), and then it gradually decreased (Figure 2). Over the temperature range of 70−85 °C, microwave treatment markedly promoted the LS-CA complex formation, and the acceleration of the complex formation was most pronounced after a large proportion of the LS granules had ruptured (70−75 °C). However, above 85 °C, LS-CA complexation decreased slightly, indicating that the binding between CA and LS had reached saturation, possibly because the higher temperatures induced thermal degradation of CA molecules and reduced the stability of the composite system. This is consistent with previous reports [30,31], which demonstrated that thermal processing accelerated the leakage and thermal degradation of polyphenols under the combined action of water and heat. Consequently, the temperature range for the preparation of LS-CA complexes was restricted to 65−85 °C in subsequent experiments, with LS without CA addition used as the control.

### 3.3. Effect of Chlorogenic Acid on the Crystal Structure of Lotus Seed Starch

Starch can crystalize in four different structural forms, i.e., A-, B-, V- and C-type structures, according to the spatial arrangement of starch chains (when A-type and B-type lattice arrangements coexist within the same particle, it is classified as C-type crystallization) [32,33]. Ungelatinized lotus seed starch at 65 °C (LS-65) exhibited C-type crystalline starch characteristics (strong diffraction peaks at 2θ values of 15°, 17°, 18° and 23° and a weak single peak at 20°; Figure 3), corresponding to a degree of crystallinity of 28.6%. With an increasing temperature, the degree of structural order gradually decreased to a minimum degree of crystallinity of 19.7% at 85 °C (LS-CA-85) [34,35], and the crystal type changed to a B-type structure (single weak diffraction peak at 17°).

The crystallographic transition is directly related to the number of bound water molecules between the starch helical units. A-type crystals are monoclinic systems with a dense arrangement of double helices, which can retain only eight bound water molecules per unit cell. In contrast, B-type crystals are hexagonal systems with a less-ordered crystal structure, which can form a unique helical inner cavity, with a repeat unit of six glucose residues, and can bind 36 water molecule ligands per unit hexagonal cell [36]. During microwave treatment, the starch molecules are mobilized under the action of electromagnetic fields; then, the free water molecules that were previously on the outer side of the helical unit enter into and participate in the reorganization of the lattice structure, mediating the transformation of the A-type structure to a B-type structure [37,38]. At higher temperatures, the lattice structure of starch molecules largely breaks down, dispersing most of the starch side chains into the surrounding medium to form a gel, allowing the hexagonal crystalline arrangement with a higher unit water content to dominate the residual crystalline starch. However, such structures have severe lattice distortion and are macroscopically closer to the sub-crystalline and amorphous intermediate states. Therefore, only a single 17° (2θ) B-type broad peak was observed for LS-80 and -85 and for LS-CA-80 and -85.

The relative crystallinity of LS-CA was lower than that of LS at the same degree of gelatinization (27.3% at 65 °C). LS started to show extensive structural disruption at 75 °C (Figure 3, LS-75; i.e., gelatinization started at ~75 °C), whereas LS-CA showed the same crystallization disruption at about 70 °C (LS-CA-70; i.e., gelatinization started at ~70 °C). The plasticizing effect of CA molecules appears to weaken the interactions between starch molecules, reducing the minimum energy required for water molecules to enter the lattice structure and facilitating the crystallographic transition. This is consistent with the findings of Section 3.1, which stated that the presence of CA improved the efficiency of microwave energy–thermal energy conversion in starch grains, thereby accelerating starch gelatinization and disruption of the starch crystals. Moreover, even when the starch was completely gelatinized (LS-85 and LS-CA-85), the relative crystallinity was different, at 19.7 and 17.3%, respectively. This phenomenon was attributable mainly to the presence of a non-uniformly ordered starch structure on a sub-microscopic scale in LS. Generally, macromolecular chains in an amorphous state on a macroscopic scale have localized ordered crystalline microregions with different orientations (malleable molecular chains aggregate to form a localized parallel arrangement). Although such chain-folded sheet crystals are not strictly long-range ordered structures, the same X-ray scattering interference is induced. However, the scattering by such non-uniform microcrystals was attenuated in LS-CA, suggesting that CA molecules inhibited the self-assembly of starch chains on the sub-microscopic scale [39].

### 3.4. Effect of Chlorogenic Acid on the Thermal Stability of Lotus Seed Starch

Thermogravimetric (TG) analysis is an effective method to assess the thermal behavior of starch, which indicates how the physical state of the starch varies with temperature [40]. The thermogravimetric (TGA; variation curve of the quality change in starch with temperature during the heating process) and differential thermogravimetric (DTG; derivative curves of the TGA for more accurate determination of the different mass loss stages) curves of LS and LS-CA with different degrees of gelatinization were determined by heating the starch samples from 30 to 800 °C (Figure 4). The TG curves of all the samples were very similar, whereas the DTG curves showed some variation. Essentially, the thermal degradation of LS proceeded in three stages. The loss of mass in the first stage (30–200 °C) mainly resulted from the volatilization of water [41]. Although the starch crystalline formed shifts from a C- to B-type structure, going from LS-65 to LS-85, and contained more bound water molecules with the increasing degree of gelatinization, the decreased crystallinity reduced the total bound water content (Table 2). Therefore, the mass loss from water volatilization decreased slightly (Figure 4) as the degree of starch gelatinization increased. Notably, the decrease in water loss was observed from LS-75 and LS-CA-70, apparently because the starting point of starch gelatinization was at a lower temperature in the presence of CA (as discussed above).

The second and main stage of mass loss, from 200–550 °C, resulted from the degradation and decomposition of the more stable structures in the starch (e.g., starch chain-chain structure) [42]. At this stage, the mass loss from LS was higher than that from LS-CA at the same degree of gelatinization (as indicated by the smaller negative peak in the DTG curves), indicating that the presence of CA stabilized the starch molecules, thereby reducing starch degradation and the mass loss.

The third stage of mass loss occurred from 550–800 °C. The starch in this temperature range would be almost completely degraded into small molecules, pyrolyzed [43] or carbonized, and the difference in mass loss between LS and LS-CA was small. However, the greatest mass loss was observed from LS-80 and LS-CA-75, consistent with the temperature point of maximum CA increase rate (Figure 2) and therefore the point of maximum starch–CA complex formation, and which would be the intermediate state of starch structure transformation from an ordered to a disordered state. During starch gelatinization, the ungelatinized starch contained a relatively high proportion of crystalline structure. When gelatinization was complete, the starch chains were released from the crystals and underwent collisional rearrangement, resulting in a non-uniformly ordered structure. However, when the starch structure was in the intermediate state, between the ordered and disordered states, the starch molecules were neither completely free from the residual crystal structure nor were they able to combine and rearrange with other chain segments in the system, which resulted in the starch structure at the temperature of maximum CA content being less ordered and thermal degradation and carbonization being more extensive.

### 3.5. Effect of Chlorogenic Acid on the Swelling Rate and Solubility of Lotus Seed Starch

An increase in gelatinization results in the breakage of hydrogen bonds between the starch molecules and the crystalline region of the starch granules changes from a compact to a looser state, accompanied by the absorption of many water molecules, resulting in extensive swelling of the starch granules [44]. After microwave treatment, the swelling of LS and LS-CA was highly consistent with the degree of gelatinization (Table 3). The granular structure of starch is a macroscopic representation of the arrangement of starch building units. When the starch granules are gelatinized, their internal structural domains no longer contain a dense core, the starch molecules are less dense and more porous, and water molecules can easily access the interior [26]. This results in a gradual expansion of the voids between the starch chains, which results in a sharp increase in the degree of swelling. However, the degree of swelling of LS-CA-70 and LS-CA-75 was significantly higher than that of LS-70 and LS-75, indicating that swelling occurred at a lower temperature, in the presence of CA. This was attributable to gelatinization at a lower temperature during microwave heating in the presence of CA, which accelerated starch granule swelling and disruption, so LS-CA near the gelatinization point had greater structural flexibility.

After gelatinization, the linear starch chains were able to diffuse freely, but interchain hydrogen bonds were very easily formed between them [45,46], leading to the reconnection of free starch molecules to form aggregates, thereby reducing their solubility. Therefore, solubility determination was used to quantify the free starch chain segments in aqueous solution. An increase in the degree of gelatinization resulted in unfolding of the starch chains, so that many hydrophilic hydroxyl groups would be exposed to the bulk aqueous medium, increasing their solubility (Table 3). The presence of CA effectively reduced the solubility of the LS-CA complexes, which may be explained by CA molecules competing with water molecules for binding to starch chains, reducing the number of hydrogen bonds between water and starch molecules. In addition, CA molecules may induce aggregation of free starch chains by hydrogen bonding to two chains, effectively cross-linking them and reducing their freedom of movement.

### 3.6. Effect of Chlorogenic Acid on the Digestive Properties of Lotus Seed Starch

The digestion curves of LS and LS-CA and the CA release at the corresponding digestion levels were determined (Figure 5A–C). The crystalline structure of starch is a key factor influencing the rate and extent of starch digestion; the higher the crystallinity, the more resistant the starch to digestion [47,48]. The decrease in starch crystallinity with the increase in the degree of gelatinization resulted in an increase in the digestibility of LS and LS-CA (Figure 5A,B). The final percentage hydrolysis increased from 47 to 73% for LS and from 43 to 70% for LS-CA and to 65 and 85 °C after microwave treatment, respectively, and at any treatment temperature, LS, with lower crystallinity, was less resistant to digestion.

The digestive properties of starch–polyphenol complexes are closely related to the amylase-inhibitory activity of the polyphenols; hydrolysis of the starch releases the complexed polyphenol, which increasingly inhibits the action of the amylase, slowing the rate of starch hydrolysis [49,50]. At the lowest degree of CA complexation (LS-CA-65), the rate of CA release was the lowest (Figure 5C); correspondingly, the rate of CA release was the highest at the highest degree of complexation (LS-CA-85). LS-CA-85 had the highest CA content, resulting in the fastest rate of release as the starch was hydrolyzed.

CA may inhibit amylase action in two ways, direct inhibition by binding to the enzyme, or by binding to starch molecules, which could hinder their binding to the enzyme active-site. In addition, a previous report suggested that, in the presence of polyphenols, starch molecules form V-type complexes (RS5) with the polyphenols, through hydrophobic interactions within the helical cavity [51,52]. V-type complexes are relatively structurally stable, so they have some anti-digestive properties.

The digestion experiments also revealed the distribution between rapidly digested starch (RDS; hydrolyzed within 20 min), slowly digested starch (SDS; hydrolyzed between 20 and 120 min) and resistant starch (RS; undigested residue after 120 min; Figure 5D). In both LS and LS-CA, increasing microwave treatment temperatures resulted in markedly less RS, a small increase in SDS and markedly more RDS, i.e., microwave treatment increased the digestibility/extent of amylase hydrolysis of the starch. In addition, at any given treatment temperature, LS-CA samples contained more RS and SDS and less RDS (*p* < 0.05).

The factors determining the rate of amylase hydrolysis of starch are amylase diffusion into the starch matrix, substrate binding to the enzyme active-site and catalysis of the hydrolysis reaction, with amylase diffusion being the rate-limiting step in the initiation of starch hydrolysis [53]. Accordingly, the core matrix structure of starch granules is considered to be the main factor differentiating SDS and RS [54]. After microwave treatment, the original compact structure of the starch granules was disrupted, to form a loose matrix with a porous surface, greatly facilitating the diffusion of amylase into the grain and providing many more binding sites for amylase action. This structural change accounts for the conversion of RS into SDS and/or RDS during microwave treatment. However, the presence of CA significantly inhibited digestion of the starch matrix (Figure 5A,B), in agreement with the increased content of RS and SDS in LS-CA (Figure 5D). Therefore, it appeared that the inhibitory effect of CA molecules on amylase was a major factor in determining the hydrolytic behavior of starch systems. This inhibition is probably caused by the polyphenol molecules forming a complex with the amylase through intermolecular forces such as hydrogen bonding or van der Waals forces and/or the interaction of the polyphenol molecules with the enzyme site, thus inhibiting the conformational change in the enzyme molecule and making it difficult to bind to the substrate [55,56,57].

### 3.7. Effect of Chlorogenic Acid on α-Amylase Starch Hydrolysis Kinetics

In common with most phenolic compounds, CA inhibited the catalytic activity of α-amylase in a dose-dependent manner (Figure 6) [50], with a half-maximal inhibitory concentration (IC_50_) of ~2 mg/mL. This is in agreement with a previous report [58] demonstrating that the IC_50_ of CA for amylase is 1.96 mg/mL for the digestion of maize starch. The type of inhibition of α-amylase by CA was determined using the Lineweaver–Burk double-reciprocal plot (Figure 7). The intersection points of the 1/V vs. 1/S plots were all in the second quadrant, and the slope (K_m_/V_max_) and *y*-axis intercept (1/V_max_) of the fitted straight line increased with an increasing CA concentration, with the K_m_ increasing from 1.68 to 2.12 (mg.mL^−1^) and the V_max_ increasing from 0.34 to 0.25 (mg.mL^−1^.min^−1^) (Table 4). This suggested mixed competitive and non-competitive inhibition of α-amylase by CA. Competitive inhibition involves CA binding to starch molecules, or to the enzyme active site, and increasing the energy required for enzyme–substrate interaction, whereas non-competitive inhibition involves CA binding to α-amylase, away from the active site and decreasing its catalytic activity by distorting the enzyme secondary structure and the active site [55,56]. Compared to some studies, the inhibition of α-amylase activity by CA is stronger than the phenolic extracts of Blackstone (non-organic red wine), Orleans Hill (organic red wine) and green tea [59].

## 4. Conclusions

In this study, the effect of CA on the dielectric response of lotus seed starch during gelatinization by microwave treatment and the complex behavior between them were investigated. In addition, the relationship between lotus seed starch–chlorogenic acid complex formation and the digestive properties of the starch was revealed. In the complex, chlorogenic acid interfered with microwave energy transfer by increasing the “hysteresis effect” on water molecules, which improved the microwave energy–thermal energy conversion efficiency of the aqueous starch suspension. This accelerated the microwave gelatinization of the starch, accelerating the degradation of its crystalline structure. During starch gelatinization, many free starch chains were released from the starch crystals and combined with chlorogenic acid molecules to form complexes, stabilized by hydrogen bonding and hydrophobic interactions. The chlorogenic acid in the complexes was gradually released during enzymic digestion, enabling it to increasingly inhibit α-amylase action via a mixture of competitive and non-competitive inhibition. The α-Amylase inhibition further improved the already good slow-digestion property of lotus seed starch, in addition to the increased content of resistant and slowly digested starch induced by the presence of chlorogenic acid. Generally, chlorogenic acid had a significant effect on the dielectric response, physicochemical properties and digestive characteristics of starch, and microwave treatment was an effective means to promote interaction and complex formation between starch and phenolic compounds.

## Figures and Tables

**Figure 1 foods-12-02506-f001:**
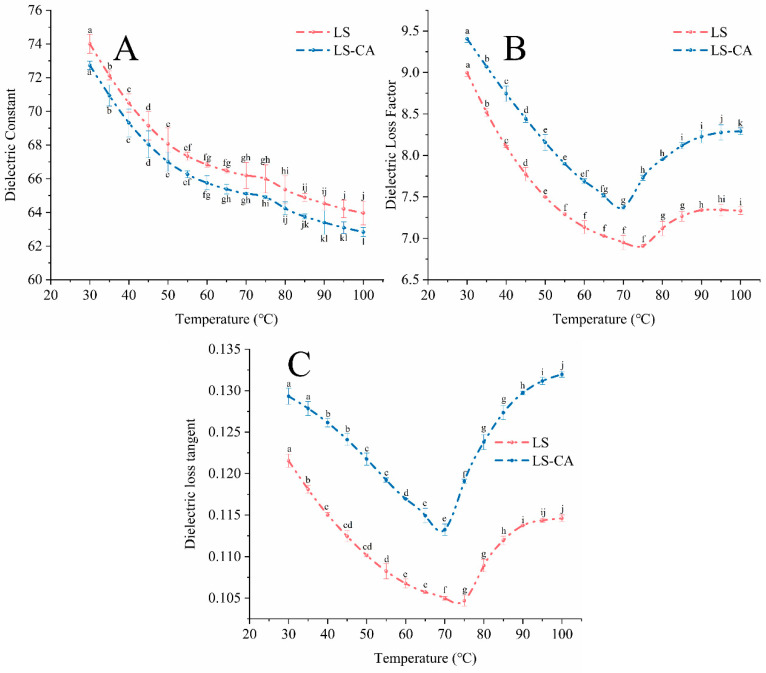
Dielectric properties of LS and LS-CA slurries at different temperatures: (**A**) dielectric constant (ε′, the ability of a substance to absorb and store electrical energy), (**B**) dielectric loss factor (ε″, the ability of a substance to convert electrical energy into thermal energy) and (**C**) dielectric loss tangent (tanδε; ε″/ε′, the degree to which a substance converts absorbed electromagnetic energy into thermal energy). Different lowercase letters indicate significant differences (*p* < 0.05).

**Figure 2 foods-12-02506-f002:**
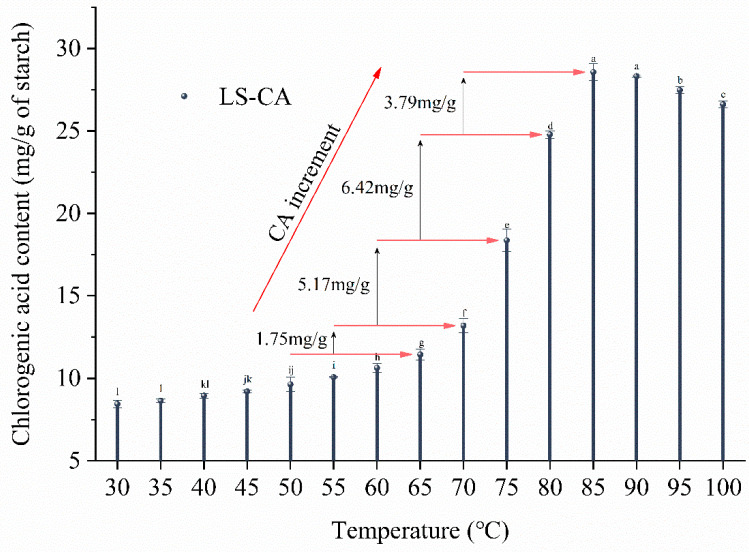
Chlorogenic acid content in LS-CA (lotus seed starch−chlorogenic acid) mixtures at different temperatures after microwave treatment. Different lowercase letters indicate significant differences (*p* < 0.05).

**Figure 3 foods-12-02506-f003:**
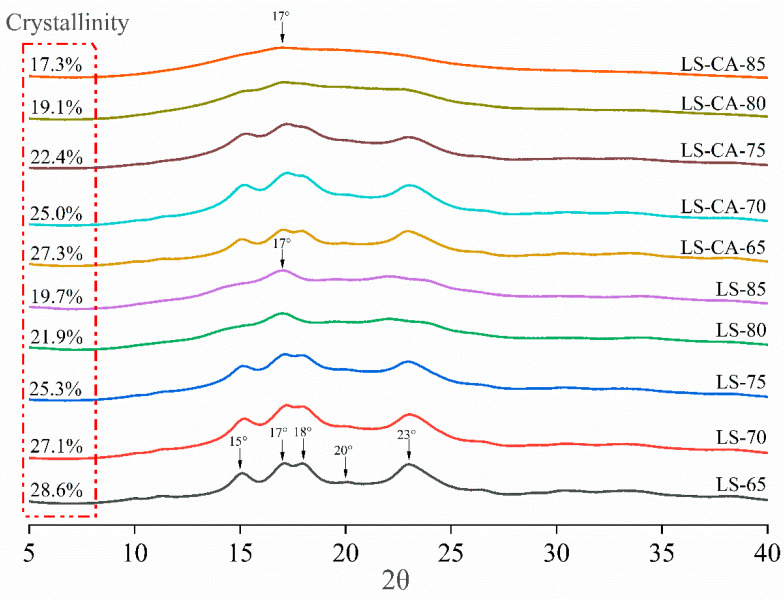
Crystal type and crystallinity of LS and LS−CA slurries treated with microwaveto final temperatures of 65 °C, 70 °C, 75 °C, 80 °C and 85 °C.

**Figure 4 foods-12-02506-f004:**
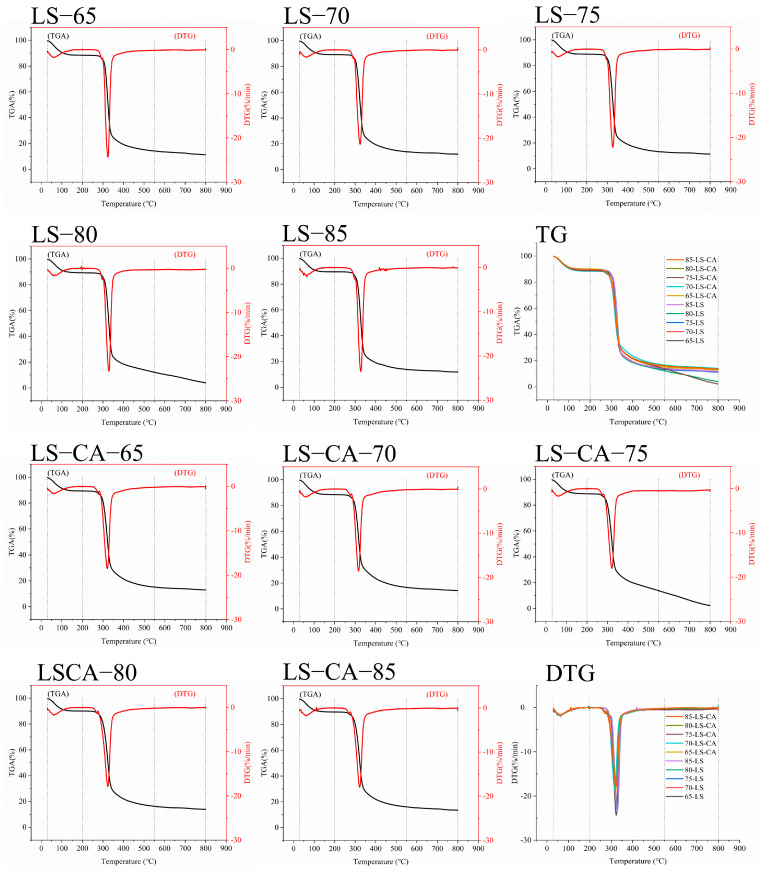
Weight loss curve of LS and LS−CA slurries treated by microwave to final temperatures of 65 °C, 70 °C, 75 °C, 80 °C and 85 °C.

**Figure 5 foods-12-02506-f005:**
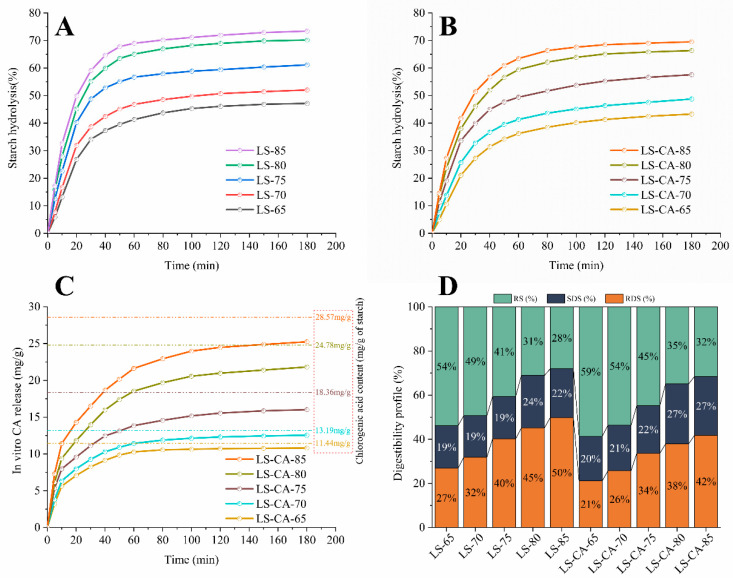
Digestion parameters of LS and LS-CA suspensions after microwave treatment to final temperatures of 65, 70, 75, 80 and 85 °C: (**A**) digestion/percentage enzymic hydrolysis curve of LS, (**B**) digestion curve of LS-CA, (**C**) CA release from LS-CA during digestion and (**D**) final composition of digested LS and LS-CA.

**Figure 6 foods-12-02506-f006:**
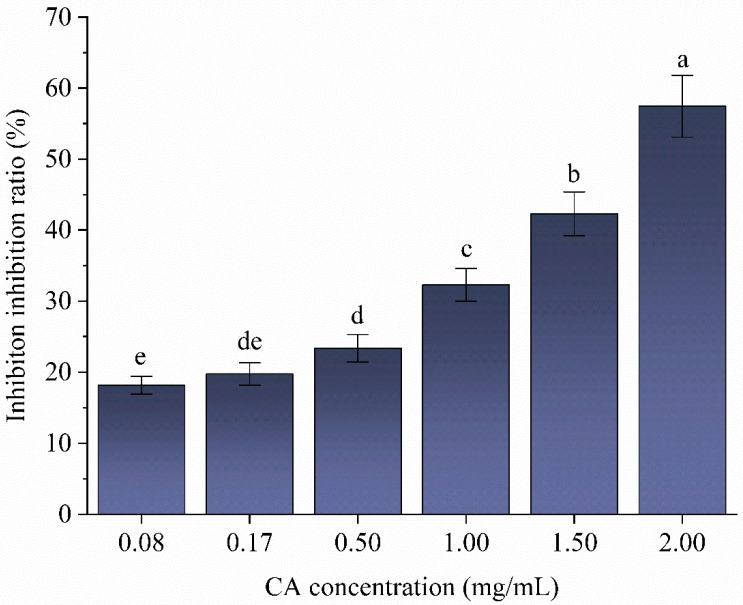
Inhibitory effect of chlorogenic acid (CA) on α-amylase activity. Different lowercase letters indicate significant differences (*p* < 0.05).

**Figure 7 foods-12-02506-f007:**
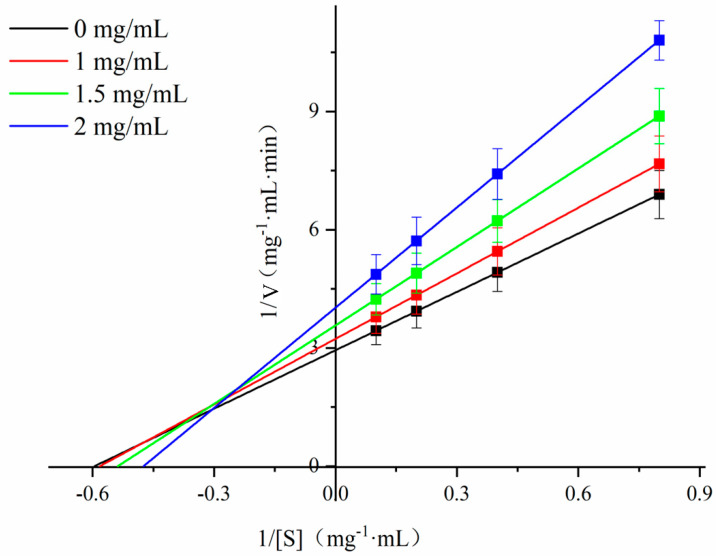
Lineweaver−Burk plots for chlorogenic acid (CA) inhibition of α-amylase.

**Table 1 foods-12-02506-t001:** Relationship between chlorogenic acid concentration and the dielectric parameters of lotus seed starch.

CA Concentration (%)	ε′	ε″	tanδε
0	76.7812 ± 0.1675 ^a^	9.5831 ± 0.0407 ^c^	0.1248 ± 0.0008 ^d^
1	75.3462 ± 0.1230 ^b^	9.7368 ± 0.0887 ^bc^	0.1292 ± 0.0012 ^c^
3	75.1475 ± 0.1191 ^bc^	9.7785 ± 0.0611 ^ab^	0.1301 ± 0.0008 ^bc^
5	74.9954 ± 0.0404 ^cd^	9.8868 ± 0.0403 ^ab^	0.1318 ± 0.0005 ^ab^
7	74.8238 ± 0.0299 ^d^	9.9050 ± 0.0526 ^ab^	0.1324 ± 0.0007 ^a^
9	74.7776 ± 0.2069 ^d^	9.9354 ± 0.1779 ^a^	0.1329 ± 0.0020 ^a^

Different lowercase letters in same column indicate significant differences (*p* < 0.05) (mean ± SD, *n* = 3).

**Table 2 foods-12-02506-t002:** Weight loss distribution of LS and LS-CA slurries treated by microwave to final temperatures of 65 °C, 70 °C, 75 °C, 80 °C and 85 °C.

Samples	30–200 °C Weight Loss (%)	200–550 °C Weight Loss (%)	550–800 °C Weight Loss (%)
65-LS	−11.38	−74.25	−2.89
70-LS	−10.54	−75.33	−1.94
75-LS	−11.03	−75.51	−1.87
80-LS	−10.41	−77.21	−10.04
85-LS	−10.24	−75.65	−2.04
65-LS-CA	−10.43	−74.15	−2.18
70-LS-CA	−11.28	−71.78	−2.49
75-LS-CA	−10.79	−75.11	−11.55
80-LS-CA	−9.75	−73.81	−2.39
85-LS-CA	−9.98	−73.68	−2.61

**Table 3 foods-12-02506-t003:** Swelling power and solubility of LS and LS-CA suspensions, heated by microwave treatment, to final temperatures of 65, 70, 75, 80 and 85 °C.

Samples	Swelling Power (SP)	Solubility (%)
LS-65	2.083 ± 0.037 ^h^	0.093 ± 0.018 ^f^
LS-70	2.571 ± 0.015 ^g^	0.152 ± 0.017 ^def^
LS-75	3.617 ± 0.076 ^f^	0.216 ± 0.029 ^d^
LS-80	5.832 ± 0.269 ^c^	0.360 ± 0.061 ^bc^
LS-85	7.146 ± 0.148 ^a^	0.459 ± 0.056 ^a^
LS-CA-65	2.034 ± 0.129 ^h^	0.081 ± 0.011 ^f^
LS-CA-70	2.751 ± 0.083 ^g^	0.137 ± 0.023 ^ef^
LS-CA-75	4.219 ± 0.120 ^e^	0.192 ± 0.037 ^de^
LS-CA-80	5.507 ± 0.274 ^d^	0.314 ± 0.040 ^c^
LS-CA-85	6.403 ± 0.258 ^b^	0.411 ± 0.059 ^ab^

Different lowercase letters in same column indicate significant differences (*p* < 0.05) (mean ± SD, *n* = 3).

**Table 4 foods-12-02506-t004:** Effect of chlorogenic acid on kinetic parameters of α-amylase.

CA (mg·mL^−1^)	Km (mg·mL^−1^)	Vmax (mg·mL^−1^·min^−1^)
0.00	1.68	0.34
1.00	1.72	0.31
1.50	1.86	0.28
2.00	2.12	0.25

## Data Availability

The data in this study were available from the following sources the corresponding authors. These data are not publicly available due to the requirement to fund research research projects.

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
