# Peer review of "Microwave-Induced Behavior and Digestive Properties of the Lotus Seed Starch—Chlorogenic Acid Complex"

_foods, 2023, doi:10.3390/foods12132506_

Round 1

Reviewer 1 Report

This research investigated the microwave induction of complexes between lotus seed starch and chlorogenic acid, and evaluated the properties of the resulting products. Overall, this manuscript is presented in a well-organized manner. It is apparent that this article is impressive and presents minimal significant issues that need to be addressed. In my opinion, the article could be further enhanced by addressing the following points, which would contribute to its overall comprehensiveness.

Line 31-32: I have some doubts regarding the frequency and wavelength of the microwave stated here. Since some references specify different frequencies and wavelengths for microwaves (ranging from 300MHz to 300GHz and wavelengths from 1mm to 1m), please double-check the accuracy and consider referring to the appropriate reference.

Line 104: lowercase “lotus seed”

There are several sections in the discussion that lack references. In my opinion, while the authors were able to discuss the results effectively, the lack of relevant research or references to support their statements makes them unreliable. The following sections are examples of content that lack references: Line 332-369, 377-386, and 391-402.

Line 443: (Wang et al., 2014)

Line 457: (Sun et al., 2016)

Line 462: Vmax decreasing from 0.34 to 0.25?

 The English quality of this article is satisfactory.

Author Response

Thank you for your comments concerning our manuscript entitled “Microwave induced behavior and digestive properties of the lotus seed starch-chlorogenic acid complex”. Those comments are all valuable and very helpful for revising and improving our paper, as well as the important guiding significance to our research. We have studied comments carefully and have made correction accordingly which we hope meet with request. Revised portions are marked in red in the paper. The main corrections in the paper and the responds to the reviewer’s comments are listed as following:

Responds to the reviewer’s comments:

Reviewer 1

Q1: Line 31-32: I have some doubts regarding the frequency and wavelength of the microwave stated here. Since some references specify different frequencies and wavelengths for microwaves (ranging from 300MHz to 300GHz and wavelengths from 1mm to 1m), please double-check the accuracy and consider referring to the appropriate reference.

A: The reviewer's comments have been taken into accounts, and necessary modifications have been made to the corresponding section. (Line 31-34)

Q2: Line 104: lowercase “lotus seed”

A: The reviewer's comments have been taken into accounts, and necessary modifications have been made to the corresponding section. (Line 114)

Q3: There are several sections in the discussion that lack references. In my opinion, while the authors were able to discuss the results effectively, the lack of relevant research or references to support their statements makes them unreliable. The following sections are examples of content that lack references: Line 332-369, 377-386, and 391-402, Line 443: (Wang et al., 2014), Line 457: (Sun et al., 2016).

A: The reviewer's comments have been taken into account and relevant references have been added. (Line 347, 362, 367)

Q4: Line 462: Vmax decreasing from 0.34 to 0.25?

A: Vmax is the reciprocal of 1/V, so its value drops from 0.34 to 0.25 (i.e. the reaction rate decreases gradually). (Line 484)

We appreciate for Editors and Reviewers’ work earnestly, and hope that the correction will meet with your approval. Once again, thank you very much for your comments and suggestions and we look forward to your positive response.

Reviewer 2 Report

The manuscript has investigated the effects of the microwave process on the physicochemical and digestibility properties of the lotus seed starch-chlorogenic acid complex. The topic is interesting; However, the manuscript has several problems:

1. L 23, 24; α-amylose. Check the entire manuscript.

2. Check the style of citations.

3. L 67-77 should be re-written. Also, express more about the lotus starch.

4. L 91; How was the moisture, ash, etc. measured?

5. L 221; gelatinization.

6. Figure 1, the figures should be presented in the same size.

7. L 294-304; What about C-type crystals? As mentioned, the lotus seed starch exhibited a C-type crystalline structure.

The manuscript should be edited by a professional English editor.

Author Response

Dear Editors and Reviewers:

Thank you for your comments concerning our manuscript entitled “Microwave induced behavior and digestive properties of the lotus seed starch-chlorogenic acid complex”. Those comments are all valuable and very helpful for revising and improving our paper, as well as the important guiding significance to our research. We have studied comments carefully and have made correction accordingly which we hope meet with request. Revised portions are marked in red in the paper. The main corrections in the paper and the responds to the reviewer’s comments are listed as following:

Responds to the reviewer’s comments:

-Reviewer 2

Q1. L 23, 24; α-amylose. Check the entire manuscript.

A: The reviewer's comments have been taken into account and revised accordingly. (Line 23-24)

Q2. Check the style of citations.

A: The reviewer's comments have been taken into account and revised accordingly.

Q3. L 67-77 should be re-written. Also, express more about the lotus starch.

A: The reviewer's comments have been taken into account and revised accordingly. (Line 65-87)

Q4. L 91; How was the moisture, ash, etc. measured?

A: The moisture and ash content of lotus seed starch are determined by the Chinese national standard method. (Line 101-102)

Q5. L 221; gelatinization.

A: The reviewer's comments have been taken into account and revised accordingly. (Line 230)

Q6. Figure 1, the figures should be presented in the same size.

A: The reviewer's comments have been taken into account and revised accordingly. (Line 249-250)

Q7. L 294-304; What about C-type crystals? As mentioned, the lotus seed starch exhibited a C-type crystalline structure.

A: C- type crystals are formed by mixing A- and B- type crystals, and there is a process of interconversion between the crystals. When the A- or B-type crystals disappear, the remaining crystals assume the state of another type of crystal. (Line 290-292, 306-310)

We appreciate for Editors and Reviewers’ work earnestly, and hope that the correction will meet with your approval. Once again, thank you very much for your comments and suggestions and we look forward to your positive response.

Yours sincerely,

Xu Lu

Reviewer 3 Report

Dear authors,

Please consider the following suggestions.

L55-56 - Which starch? You mean the one isolated from lotus? Which type of phenolics are mostly susceptible to complex with starch?

L77 - changing such properties to what? Easily digestible or not?

l58-l59 - Is that good or bad? Write in the text.

L114 - What is the relative degree of complexation?

L122 - Why did the authors use gallic acid as standard instead of chlorogenic acid? A previous report from you (Jiang et al. Food hydrocolloids,v144, November 2023, 108925) used chlorogenic acid as standard. How the complexation was calculated? Did the authors used a control sample (without chlorogenic acid)?

L168-L170 - I recommend the authors remove this part and in the rewrite the title of 2.4 section as ''Determination of relative degree of complexation via the quantification of chlorogenic acid with the Total Phenolic Content assay''. Avoid writing similarities with your previous work published (Jiang et al. Int Journal of Biol Macromolecules, v.191, 30 Nov 2021, pg 474-482).

L337/Figure 4 - Explain the differences and what do they mean.  Overlap the curves (starch + starch/phenolic in the same graphic) and check how significant the treatments were for TG. The authors can insert a supplementary table reporting the weight loss per treatment.

l450-451 - How chlorogenic acid inhibited alpha amylase activity? Discuss in the text.

Topic 3.7/Table 3 - This finding is interesting and important! The authors must mention that inhibition of alpha amylase is important in foods to control the resistance to insulin. Compare this work with literature: how much is that inhibition compared with phenolic extracts?

Table 3 - How many kinetics were done? Are these data resulted from triplicate? If yes, insert standard deviation.

Author Response

Dear Editors and Reviewers:

Thank you for your comments concerning our manuscript entitled “Microwave induced behavior and digestive properties of the lotus seed starch-chlorogenic acid complex”. Those comments are all valuable and very helpful for revising and improving our paper, as well as the important guiding significance to our research. We have studied comments carefully and have made correction accordingly which we hope meet with request. Revised portions are marked in red in the paper. The main corrections in the paper and the responds to the reviewer’s comments are listed as following:

Responds to the reviewer’s comments:

-Reviewer 3

Q1. L55-56 - Which starch? You mean the one isolated from lotus? Which type of phenolics are mostly susceptible to complex with starch?

A: This part of "starch" for a variety of starch collectively, such as potato starch (https://doi.org/10.1016/j.ijbiomac.2020.10.209), rice starch (https://doi.org/10.1016/j.lwt.2020.109227) and lotus seed starch (https://doi.org/10.1016/j.foodchem.2019.124992) can be compounded with starch.The difficulty of complexation is influenced by various factors (size of polyphenol molecules, spatial resistance, number of phenolic hydroxyl groups, etc.), for example, phenols with small molecular size (https://doi.org/10.1016/j.tifs.2015.02.003) can form hydrogen bonds with starch side chain hydroxyl groups , and enter the cavities of starch helical chains under the "pull" of hydrogen bonds and hydrophobic interactions to form V-shaped complexes; while plant polyphenols with large spatial resistance are limited by the size of starch hydrophobic cavities and are unable to enter the starch helical cavities through hydrophobic interactions, so the formation of hydrogen bonds has an important role in maintaining the stable binding process of these plant polyphenols to starch (https://doi.org/10.1016/j.tifs.2021.04.032). At the same time, the number and strength of hydrogen bonds formed between the two are influenced by the number of phenolic hydroxyl groups of plant polyphenols, the higher the number of phenolic hydroxyl groups of plant polyphenols, the greater the number of hydrogen bonds formed between them and starch, and the greater the bond strength between them. (https://doi.org/10.1016/j.foodhyd.2019.105409).

Q2. L77 - changing such properties to what? Easily digestible or not?

A: The modified starch has anti-digestive properties and the sentence has been rewritten in the paper. (Line 83-87)

Q3. l58-l59 - Is that good or bad? Write in the text.

A: The reviewer's comments have been taken into accounts and the sentence has been rewritten. (Line 55-60)

Q4. L114 - What is the relative degree of complexation?

A: The relative degree of complexation is the total polyphenol content in the lotus seed starch system after the removal of free chlorogenic acid, and the title has been rewritten. (Line 124-125)

Q5. L122 - Why did the authors use gallic acid as standard instead of chlorogenic acid? A previous report from you (Jiang et al. Food hydrocolloids,v144, November 2023, 108925) used chlorogenic acid as standard. How the complexation was calculated? Did the authors used a control sample (without chlorogenic acid)?

A: In this experiment, the polyphenol content in the material was determined by the oxidation reaction between forintanol and the electrophilic sites in the polyphenol molecule. Polyphenols contain several phenolic hydroxyl groups (-OH), which have strong redox ability, and easily oxidize with substances of strong oxidizing properties to produce colored products. Whereas, the easily accessible gallic acid has similar structure and properties to chlorogenic acid, and is a good substitute for standards in simple intergroup qualitative analysis experiments. Gallic acid is widely used for the determination of total phenolic content in materials, which was cited in a large number of research articles on starch polyphenol complex systems. For comparison purposes, the easily accessible gallic acid was chosen as a proxy. (https://doi.org/10.1016/j.foodhyd.2022.108280; https://doi.org/10.1016/j.foodchem.2006.06.051; https://doi.org/10.1016/j.ijbiomac.2023.125457; https://doi.org/10.1016/j.foodhyd.2021.106966).

Q6. L168-L170 - I recommend the authors remove this part and in the rewrite the title of 2.4 section as ''Determination of relative degree of complexation via the quantification of chlorogenic acid with the Total Phenolic Content assay''. Avoid writing similarities with your previous work published (Jiang et al. Int Journal of Biol Macromolecules, v.191, 30 Nov 2021, pg 474-482).

A: The reviewer's comments have been taken into account and revised accordingly.  (Line 124-125)

Q7. L337/Figure 4 - Explain the differences and what do they mean.  Overlap the curves (starch + starch/phenolic in the same graphic) and check how significant the treatments were for TG. The authors can insert a supplementary table reporting the weight loss per treatment.

A: The reviewers' comments have been taken into account and the corresponding graphics and tables have been added. (Line 340-343, 355-359)

Q8. l450-451 - How chlorogenic acid inhibited alpha amylase activity? Discuss in the text.

A: The reviewer's comments have been taken into account and added accordingly. (Line 462-466, 477-481)

Q9. Topic 3.7/Table 3 - This finding is interesting and important! The authors must mention that inhibition of alpha amylase is important in foods to control the resistance to insulin. Compare this work with literature: how much is that inhibition compared with phenolic extracts?

A: The reviewer's comments have been taken into account and added accordingly. (Line 481-483)

Q10. Table 3 - How many kinetics were done? Are these data resulted from triplicate? If yes, insert standard deviation.

A: Table 3 (Effect of chlorogenic acid on kinetic parameters of α-amylase) is obtained by fitting the mean values of the inhibition of α-amylase activity by chlorogenic acid, so there is no standard deviation.

We appreciate for Editors and Reviewers’ work earnestly, and hope that the correction will meet with your approval. Once again, thank you very much for your comments and suggestions and we look forward to your positive response.

Yours sincerely,

Xu Lu

Round 2

Reviewer 1 Report

I'm satisfied with the revised version.